# Insecticide resistance of Miami-Dade *Culex quinquefasciatus* populations and initial field efficacy of a new resistance-breaking adulticide formulation

Isik Unlu[1]*, Eva A. Buckner[2], Johanna Medina[1], Chalmers Vasquez[1], Aimee Cabrera[1], Ana L. Romero-Weaver[2], Daviela Ramirez[2¤], Natalie L. Kendziorski[2], Kyle J. Kosinski[2], T. J. Fedirko[2], Leigh Ketelsen[2], Chelsea Dorsainvil[2], Alden S. Estep[3]*

**1** Miami-Dade County Mosquito Control Division, Miami, Florida, United States of America, **2** Department of Entomology and Nematology, Florida Medical Entomology Laboratory, Institute of Food and Agricultural Sciences, University of Florida, Vero Beach, Florida, United States of America, **3** Center for Medical, Agricultural and Veterinary Entomology, Agricultural Research Service, United States Department of Agriculture, Gainesville, Florida, United States of America

¤ Current address: Highland Agricultural Solutions, Mulberry, Florida, United States of America
* isik.unlu@miamidade.gov (IU); alden.estep@usda.gov (ASE)

**Data Availability Statement:** All relevant data are within the manuscript and the Supporting files.

## Abstract

Sporadic outbreaks of human cases of West Nile virus (WNV), primarily vectored by *Culex quinquefasciatus* Say in suburban and urban areas, have been reported since introduction of the virus into Florida in 2001. Miami-Dade County, Florida is part of one of the largest metropolitan areas in the United States, supports *Cx. quinquefasciatus* year-round, and recently experienced over 60 human cases of WNV during one outbreak. To facilitate more effective integrated vector management and public health protection, we used the Centers for Disease Control and Prevention (CDC) bottle bioassay method to evaluate the susceptibility of adult *Cx. quinquefasciatus* collected from 29 locations throughout Miami-Dade County to pyrethroid and organophosphate adulticide active ingredients (AIs) used by Miami-Dade County Mosquito Control. We also determined the frequency of the 1014 knockdown resistance (*kdr*) mutation for *Cx. quinquefasciatus* from a subset of 17 locations. We detected resistance to two pyrethroid AIs in all tested locations (permethrin: 27 locations, deltamethrin: 28 locations). The 1014F allele was widely distributed throughout all 17 locations sampled; however, 29.4% of these locations lacked 1014F homozygotes even though phenotypic pyrethroid resistance was present. Organophosphate resistance was more variable; 20.7% of the locations tested were susceptible to malathion, and 33.3% of the populations were susceptible to naled. We subsequently conducted a field trial of ReMoa Tri, a recently approved multiple AI adulticide formulation labelled for resistant mosquitoes, against a mixed location field population of Miami-Dade *Cx. quinquefasciatus*. Average 24-hr mortality was 65.1 ± 7.2% and 48-hr mortality increased to 85.3 ± 9.1%, indicating good control of these resistant *Cx. quinquefasciatus*. This current study shows that insecticide resistance is common in local *Cx. quinquefasciatus* but effective

**Funding:** Funding for performing the bottle bioassays was provided by the Centers for Disease Control and Prevention (contract no. NU50CK000420-04-04) and Florida Department of Health (contract no. CODQJ) to EAB. The funders had no role in study design, data collection and analysis, decision to publish, or preparation of the manuscript.

**Competing interests:** ReMoa Tri for field testing was provided by Valent Biosciences LLC under an Experimental Use Permit. Valent BioSciences LLC holds a patent (US9826742) related to this product. This does not alter our adherence to PLOS ONE policies on sharing data and materials.

options are available to maintain control during active disease transmission in Miami-Dade County.

## Introduction

Chemical insecticides have been heavily used since the 1940s and have resulted in insecticide resistance (IR) in many insects that serve as vectors of arthropod-borne viruses [1]. Insecticide resistance can develop quickly and has been reported against all major classes of active ingredients (AIs) commonly used for public health. Development of IR and the ban on DDT from the USA catalyzed the need to implement integrated pest management (IPM) and develop additional insect control methods and products. In 1972, the United States Congress initiated a program for environmental protection and proposed IPM measures intended to support responsible chemical insecticide use [1, 2]. These events encouraged the adoption of IPM-driven surveillance programs in many mosquito control abatement programs in the USA. Insecticide resistance monitoring has long been encouraged by public health agencies [3–5] as a critical element of an effective program, and testing has expanded within the USA since the 2016 Zika outbreak [6]. However, most mosquito control agencies lack a robust resistance monitoring program and lack the baseline resistance data needed to inform effective operational decision-making [7, 8].

Adulticides for mosquito control in the US have long been limited to only two modes of action, namely organophosphates and pyrethroids. Both modes of action have been used extensively causing significant selection for resistance in many mosquito populations [9]. Resistance in mosquitoes is hypothesized to be primarily from non-public health use such as agriculture and consumer home use products, which are heavily biased towards pyrethroids. Mosquitoes evolve two major types of resistance in response to selection and these are distinct physiological mechanisms of adaptation [9, 10]. Metabolic resistance is the mechanism by which the insecticide molecules are degraded by enzymes from the cytochrome P450, esterase, glutathione transferase families present in the mosquitoes. Target site insensitivity, for example, knockdown resistance (*kdr*), is a heritable genetic mutation, which reduces or prevents the binding of insecticides to receptors or sites of action within cells [11].

Miami-Dade County, Florida is home to approximately 2.7 million people with a tropical climate. *Aedes* and *Culex* vector species are widely present and suspected cases of vector-borne disease are regularly reported to the Florida Department of Health (FDOH) [12]. These disease case reports require an operational response from the Miami-Dade Mosquito Control Division [13]. While many of these reported cases are travel-associated, local transmission does occasionally occur. *Aedes aegypti*-vectored pathogens represent approximately 80% of confirmed human cases [12]. Prior IR surveillance in Miami-Dade County has demonstrated widespread pyrethroid resistance of variable intensity and the presence of characteristic *Ae. aegypti kdr* mutations [14, 15].

*Cx. quinquefasciatus* is responsible for approximately 20% mosquito vectored human cases reported in Miami-Dade [12]. It is one of the two most important West Nile virus (WNV) vector species in Florida, and large populations are present year-round in Miami-Dade County, but much less is known about IR in these *Cx. quinquefasciatus* populations compared to *Ae. aegypti* [16]. *Culex quinquefasciatus/pipiens* complex populations are often resistant to common AIs [17–23]. Mechanistically focal *kdr* resistance, identified as a single nucleotide polymorphism (SNP) resulting in an amino acid change from leucine to phenylalanine at position 1014 (1014L to 1014F or occasionally 1014S), has been described in numerous populations,

but published work indicates that enzymatic resistance often accounts for a large proportion of the observed IR phenotype in *Cx quinquefasciatus* and that these same enzymes are able to degrade other classes of AIs resulting in cross-resistance [21, 23–25]. Effective long-term control of *Cx. quinquefasciatus* requires implementation of an integrated mosquito management approach to maintain populations at acceptable levels and reduce the occasional transmission of vector borne disease. The development of new tools, including combination AI formulations, has shown promise to provide options with immediate results that remain effective even against insecticide resistant populations [26–28].

ReMoa Tri (Valent BioSciences LLC) is an adulticide containing multiple AIs with three different modes of action including one that is novel to public health. The recently EPA-approved label states that it can be used for permethrin resistant *Culex* and *Aedes* mosquitoes. The AIs are 1.5% abamectin (macrocyclic lactone), 4% fenpropathrin (Type II synthetic pyrethroid) and 1% C8910 (fatty acid). Abamectin, derived from bacterial fermentation of *Streptomyces avermitilis*, interferes with the glutamate-gated chloride channels of invertebrates, and has been used in human and animal health industries as an antiparasitic [29, 30]. The fatty acid C8910 has been documented to have repellent properties in low doses, lethal effects at higher doses, and effects the respiratory system of mosquitoes [31–33]. Previous reports indicate that the C8910 fatty acid synergizes with pyrethroids [34, 35].

Miami-Dade Mosquito Control began utilizing resistance monitoring information during the 2020 mosquito season when there were over 60 WNV human cases and *Cx. quinquefasciatus* made up 85% of 13 WNV-positive mosquito pools in the county reported by the Florida Department of Health [12]. Assessing the IR status of local Miami-Dade County *Cx. quinquefasciatus* populations and identifying effective methods to control these vector mosquitoes have become a critical priority. With this essential information, operational treatments can be chosen to be as efficient as possible to protect public health. Without this knowledge, inefficient adult mosquito treatment may occur and can lead to failure of adult mosquito management during a mosquito-borne disease outbreak.

## Methods

### Site description and mosquito collection

Miami-Dade County is a county located in the southeastern part of Florida and much of the county is heavily urbanized. It also includes two federal preserves, the Everglades National Park and Biscayne Bay Aquatic Preserves. Our study focused primarily on urbanized, populated areas located in the northern part of the county, but we also sampled areas in the less dense and more agricultural southwestern portion of the county (S1 Table). We chose areas that are treated with adulticides year-round. All collection locations were within the jurisdiction of the Miami-Dade County governance domain and made by professional mosquito control personnel. Entomological surveys and collections made on private lands or residences were conducted with oral or written consent from residents. No specific permits were required for the mosquito collections. These studies did not involve endangered or protected species.

Oviposition habitats consisted of 11-L black buckets containing 8 L of tap water with 25 g of grass cutting and 2 weepholes just above the water level to maintain a constant level. Buckets were placed in locations for 7 days to allow sufficient time for egg raft oviposition.

Data on the location, deployment, service, and retirement of each *Culex* oviposition bucket was managed in an oviposition layer of the Miami-Dade Mosquito Control Geographic Information System. Individual *Culex* egg rafts laid in the oviposition bucket were transferred with a paintbrush onto moistened filter paper in a 30mm petri dish. Each small Petri dish was wrapped with parafilm and labeled with the corresponding address (site trap ID name),

collection date, and raft number. *Culex spp.* egg rafts were collected in the morning and shipped to the University of Florida, Institute of Food and Agricultural Sciences, Florida Medical Entomology Laboratory (UF/IFAS FMEL) located in Vero Beach, FL. Mosquitoes for the field spray portion of the study were collected as rafts from residential neighborhoods in Doral and Miami Beach in the Eastern portion of the county and reared as below.

## Mosquito rearing and insecticide susceptibility assays

Adult mosquito rearing took place within walk-in insectaries, maintained at 27 ± 2˚C and 50–60% relative humidity. Each *Culex* egg raft was placed in a separate enamel rearing pan containing 2-L of distilled water and larval diet of 1:1 by weight of lactalbumin and Brewer's yeast. After first instar larvae were observed in a pan, the larval diet was changed to ground pig food and added *ad libitum*. *Culex* mosquitoes were identified to species as third larval instars using a morphological key [36]. Pupae were transferred from rearing pans to 30.5 × 30.5 × 30.5 cm cages (Bioquip). Emerged adults were provided with 10% sucrose saturated cotton. A colony of CMAVE *Cx. quinquefasciatus* was reared under the same conditions described above for the field populations and served as the susceptible reference strain in insecticide susceptibility assays [37].

The standard Centers for Disease Control and Prevention (CDC) bottle bioassay method was used to determine the insecticide susceptibility to two pyrethroid AIs (deltamethrin and permethrin) and two organophosphate AIs (malathion and naled). Bottle bioassays were conducted following the protocol provided by the CDC [38]. The CDC suggested AI-specific diagnostic doses and diagnostic times were used (permethrin: 43 μg/bottle & 30 minutes, deltamethrin: 0.75 μg/bottle & 60 minutes, malathion: 400 μg/bottle & 45 minutes, naled: 2.25 μg/bottle & 45 minutes), and AI stock solutions were prepared in acetone (Thermo Fisher Scientific, Waltham, MA) using technical grade AIs obtained from CDC. Each bottle bioassay performed contained four AI treated bottles and one negative control (acetone only) bottle.

Approximately 10 to 25 6-12-days-old $F_0$ adult *Cx. quinquefasciatus* mosquitoes were aspirated into each bottle and monitored for mortality with counts taken at 0, 5, 10, and 15 minutes and then every 15 minutes through two hours. Dead mosquitoes were those that could no longer fly or stand. Mosquitoes alive at the end of a bioassay were killed by freezing, and then the total number of mosquitoes in each bottle was counted to calculate the percent mortality at all time points. If the percent mortality in the negative control bottle was between 3 to 10%, mortality in treated bottles was corrected using Abbott's formula. If percent mortality in the control bottle was greater than 10%, the assay results were discarded [38, 39]. The Miami-Dade County field *Cx. quinquefasciatus* populations were considered susceptible, developing resistance, or resistant if they experienced ≥97%, 90–97%, or <90% mortality, respectively, at the DT determined for each AI. Raw bottle bioassay data for each population is available in S1 File (S1 File).

## Assessment of the 1014 knockdown resistance (*kdr*) mutation

Miami-Dade adult female *Cx. quinquefasciatus* were shipped on dry ice from UF/IFAS FMEL to United States Department of Agriculture, Agricultural Research Service, Center for Medical, Agricultural and Veterinary Entomology (USDA ARS CMAVE) for *kdr* testing. Assessment of the SNP that results in an amino acid change from the normal leucine to a phenylalanine (occasionally serine) residue at amino acid position 1014 (based on the position in the standard *Musca domestica* sodium channel) was conducted using a melt curve assay described previously where alleles are differentiated by melting temperature [23, 40].

As described in [23], individual *Cx. quinquefasciatus* (>25 per location) were homogenized in 400 µl of nuclease free water. Two controls for each allele (1014L from the CMAVE strain and 1014F organisms from an in-house Louisiana (LA) strain with known *kdr* mutations), 2 blank wells with no mosquito, and 2 wells of an artificial heterozygote created by the addition of one CMAVE and one LA strain mosquito were included in assays.

PCR reactions were assembled in 384-well plates on an Eppendorf 5750 workstation. Each 10 µl PCR reaction consisted of 5 µl of SYBRgreen Select (Thermo Fisher), 2.90 µl NFW, 0.033 µl of common 100 µM reverse primer Cxq_1014Rev, 0.033 µl of 100 µM 1014_F primer, 0.025 µl of 100 µM 1014_S primer, 0.00575 µl of 100 µM 1014_L primer, and 2 µl of mosquito homogenate [23]. Reactions were cycled on a QuantStudio6 Flex for 40 cycles using standard FAST parameters with a final melt curve phase from 60˚C to 95˚C.

Allele calls for each organism were made as described in [23] by examining each individual melt curve to determine $T_m$ peaks (~82.2 ± 0.4˚C for 1014F, ~86.0 ± 0.4˚C for 1014L). Control wells were examined to ensure assay validity by having peaks at the expected temperatures. Heterozygotes (LF) gave peaks at both $T_m$s. Frequencies for each of the three genotypes (LL, LF, FF) and the L and F alleles were calculated by dividing the specific genotype by the total number tested from each population as in [25].

Correlation analysis of mortality at the DT for permethrin and deltamethrin versus *kdr* genotype percentage was conducted in Prism 9.4.1 (Graphpad Software, LLC) using Spearman's correlation rather than Pearson's correlation as Spearman's allows non-normal data and handles monotonic relationships that may not necessarily be linear. Analysis was conducted for the 17 sites that had both *kdr* and bottle bioassay data (S1 File).

## ReMoa tri ground ULV field trial

Adulticide application was made using a Clarke Grizzly Cold Aerosol Generator (Clarke Mosquito Control Products, Roselle, IL), a truck-mounted, calibrated Ultra-Low Volume (ULV) aerosol spray system equipped with a smart flow system. Additionally, a fluorescent dye (Tinopal® OB) was added at a ratio of two g per liter of ReMoa Tri, allowing the ULV droplets collected during application to be visible using an ultra-violet microscope.

To ensure that the ReMoa Tri application was made when weather conditions met label requirements and would allow the ULV spray cloud to stay at ground level, two weather stations were placed upwind of the planned ULV spray truck path at heights of 1.5 m (Kestrel 3550 AG Pocket Weather Meter with LiNK and Vane Mount) and 9.14 m (Kestrel 5200L Pocket Weather Meter with LiNK and Vane Mount) to monitor temperature and wind speed. The field trial site consisted of three rows of three sampling stations spaced approximately 30.5 m (100 ft) apart and located at 30.5 m (100 ft), 70.0 m (200 ft), and 91.4 m (300 ft), respectively, downwind and perpendicular to the planned ULV spray truck path. Additionally, three untreated control sampling stations and an additional weather station were positioned upwind of the three-by-three treatment grid. Each sampling station consisted of a tripod equipped with a rotating impinger (Leading Edge Associates, Fletcher, NC) holding two acrylic 3 mm Teflon-coated rods for collecting droplets and two disposable mosquito cages (15.2 cm diameter x 3.8 cm) containing approximately 20 mixed Miami Beach and Doral adult female *Cx. quinquefasciatus* mosquitoes -approximately 1.5 m above the ground. Counted mosquito cages were hung and impingers were turned on immediately prior to spray application.

Application was conducted approximately 1 hour after sunrise. During the field trial, the product was applied at 0.8 ounces/acre at a flow rate of 4.8 oz/min. The truck with the ULV equipment was driven at 10 mph. The ULV sprayer was turned on 200 feet before the treatment sampling stations and turned off 200 feet after passing the treatment sampling stations.

Mosquito cages and acrylic rods were collected from each sampling station 20 minutes post-application and returned to the laboratory. The mosquito cages collected from untreated control sampling stations were stored in a separate container from those collected from treatment sampling stations to prevent contamination. Upon return to lab, mosquitoes were transferred into clean paperboard holding containers covered with mesh netting and supplied with cotton balls soaked in a 10% sucrose solution. The holding containers were held in a secure area of the lab at a temperature of approximately 75˚F and relative humidity of 65% ± 5%. Mosquito mortality in each holding container was assessed at 24- and 48-hours post-treatment.

Droplets were analyzed using a fluorescent microscope equipped with a digital camera and connected to a computer with Drop™ software. DropVision (Leading Edge Associates, Fletcher, NC) was used to calculate average droplet size as volume median diameter (VMD) and droplet density (flux) from at least one rod from each treatment and untreated control sampling station. Analysis of droplet (VMD and flux) and field spray mortality data (24 and 48 hr mortality) was conducted by ANOVA comparing the distances from the ULV spray lines implemented in Prism 9.4.1 (Graphpad Software, LLC).

## Results

### Insecticide susceptibility assays

Field collected rafts from the 29 locations were reared to adulthood in the laboratory and tested for insecticide susceptibility to permethrin, deltamethrin, malathion, and/or naled using the CDC bottle bioassay method. Up to four AIs were tested if sufficient organisms were available from a particular location. *Cx. quinquefasciatus* from the 27 tested locations were resistant to permethrin (Fig 1A, S2 Table). Permethrin exposure resulted in ≤50% mortality at the 30-minute DT for 89% of the populations evaluated, with seven of these populations experiencing ≤10% mortality. At 120 minutes of permethrin exposure, mortality was higher in all tested populations than at the DT but no population reached 100% mortality. The populations from 62$^{nd}$ Terrace and Prairie Avenue had mortality closest to 100% (S2 Table). Mortality after 120 minutes of permethrin exposure did not exceed 50% in four of the populations tested, and three of these were from the Homestead area of Miami-Dade County.

All 28 populations tested against deltamethrin were resistant at the DT (Fig 1B, S2 Table). Only the Caliph Street population exceeded 50% mortality at DT, and 43% experienced less than 10% mortality at the DT. The mortality rates for all other populations tested ranged between 10% and 50% at the DT. At 120 minutes of deltamethrin exposure, the highest mortality rate (77%) was observed in the Wynwood population (S2 Table). The average mortality rate for the other 27 locations was 25.6 ± 14.9% at 120 minutes.

CDC bottle bioassay testing of all 29 populations with malathion showed a range of susceptibility and resistance at the DT (Fig 2A, S2 Table). Twenty-four percent of the populations were susceptible, and all were found in the northeastern portion of the county. Eight populations (~28%) were classified as developing resistance and 7 of these 8 were also in the northeastern portion of the county. Forty-eight percent of Miami-Dade County *Cx. quinquefasciatus* populations were resistant to malathion with 10 of the 14 located in the southwestern portion of the county that includes Homestead, Florida City, and agricultural areas. Continuing the assay through 120 minutes resulted in complete mortality against malathion in 55% of the 29 populations. Three of the 4 populations with the lowest mortality at 120 minutes were from Homestead (S2 Table).

Susceptibility testing of 20 *Cx. quinquefasciatus* populations with naled, another organophosphate commonly used in Florida, showed a similar pattern to what was seen malathion (Fig 2B, S2 Table). The naled susceptible *Cx. quinquefasciatus* populations and those developing

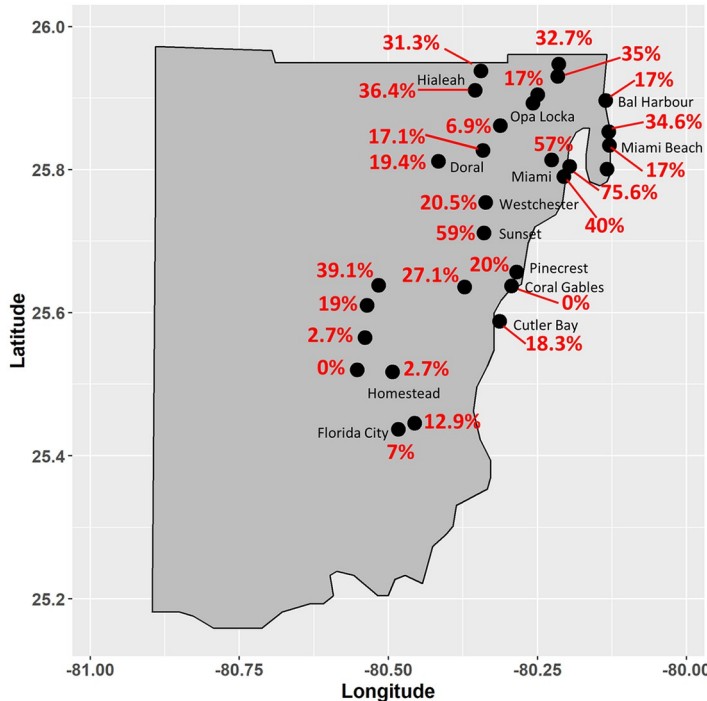

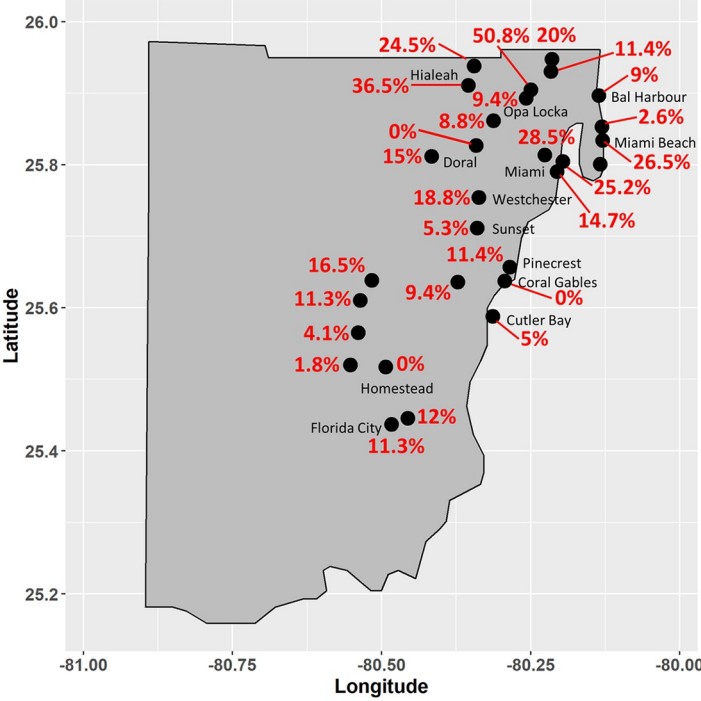

**Fig 1.** Percent mortality of Miami-Dade County *Cx. quinquefasciatus* populations at diagnostic times in CDC bottle bioassays against permethrin (A) and deltamethrin (B). Mortality percentages are colored according to outcome based on the CDC scale. Red type indicates resistance, green type indicates susceptibility, and black type indicates developing resistance according to the CDC scale [38].

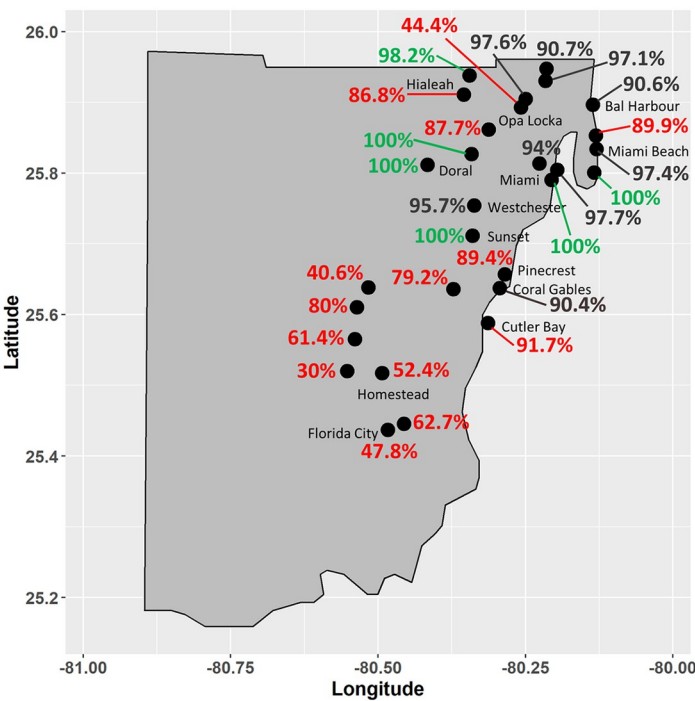

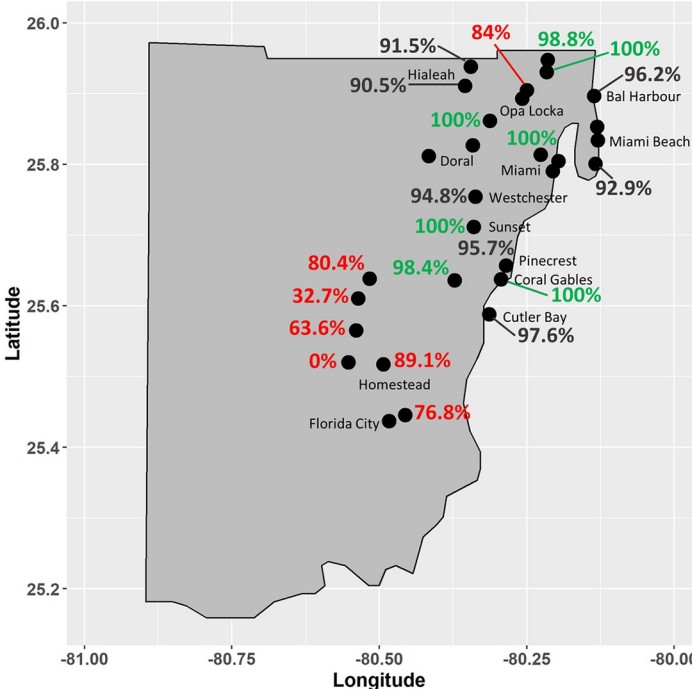

**Fig 2.** Mortality of Miami-Dade County *Cx. quinquefasciatus* populations at diagnostic times in CDC bottle bioassay malathion (A) and naled (B). Mortality percentages are colored according to outcome based on the CDC scale. Red type indicates resistance, green type indicates susceptibility, and black type indicates developing resistance according to the CDC scale [38].

resistance were collected from northern and eastern Miami-Dade County, while 6 of the 7 resistant populations were collected from western Miami-Dade County around Homestead (Fig 2B).

## Assessment of the 1014 knockdown resistance (*kdr*) mutation

Genetic testing for the presence of the 1014 leucine (1014L) to phenylalanine mutation (1014F) of 17 Miami-Dade County *Cx. quinquefasciatus* populations showed a wide variation in both frequency of the F allele and the percentage of each population homozygous (FF) for the mutation (S3 Table and Fig 3). Most of the populations had less than 10% FF, while 5 had no homozygous FF organisms. The Pinecrest population had the highest percentage of the FF genotype at 45.5%. Although FF was relatively infrequent in populations, the heterozygous LF genotype was common and averaged 50.6 ± 12.9% of each population. The LL genotype was more common than the FF genotype with 83.3% and 62.1% the highest percentages observed in the NW 42[nd] Street and W 44[th] Street populations, respectively (S3 Table). All other populations had 50% or less LL organisms.

Correlation analysis did not show significant correlation between the FF genotype and permethrin mortality (Spearman's ρ = -0.2708; p = 0.2901) but did show a moderate correlation when the same comparison was conducted with deltamethrin (Spearman's ρ = -0.5476; p = 0.0248) (S1 File).

## ReMoa tri ground ULV field trial

The wind speeds averaged approximately 4 MPH during the field trial, and there was only a minor difference between temperature at 1.5 m and 9.1 m (S4 Table). We observed no

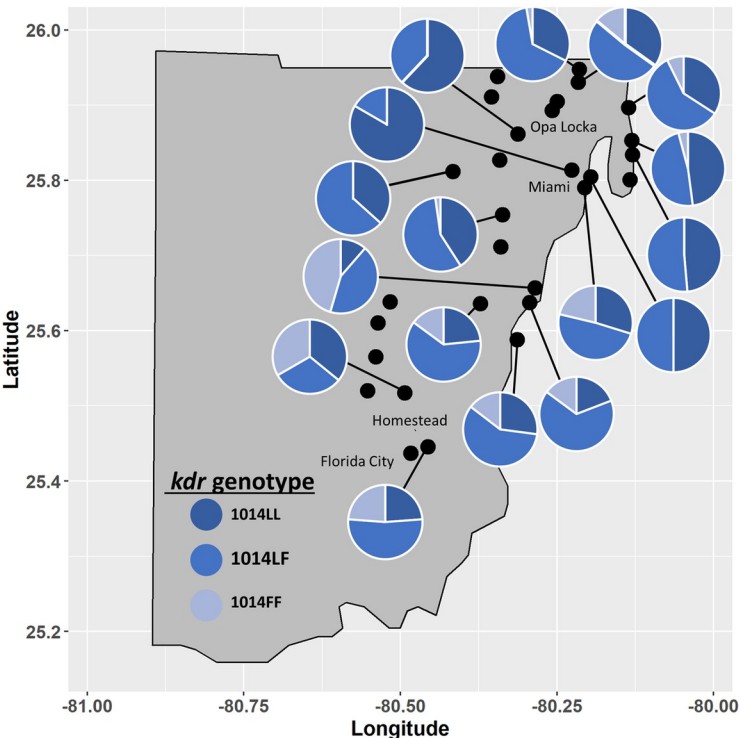

**Fig 3. Genotype percentage for the 1014 leucine to phenylalanine mutation.** Assays were conducted by melt curve assay as described in [23]. Assay controls consisted of no template, *Culex quinquefasciatus* homozygous for the wildtype (1014LL) genotype, homozygous for the *kdr* mutant (1014FF) genotype and the heterozygous (1014LF) genotype.

**Table 1. Miami-Dade County *Cx. quinquefasciatus* mortality from ReMoa Tri ground ULV field trial.**

| Distance from sprayer (meter) | 24 hr mortality (%) | 48 hr mortality (%) | total females |
|---|---|---|---|
| 30.4 | 68.8 ± 4.3 | 88.2 ± 10.7 | 131 |
| 61 | 61.7 ± 11.5 | 81.1 ± 10.8 | 120 |
| 91.4 | 64.8 ± 3.9 | 86.7 ± 7.7 | 139 |
| Avg. mortality | **65.1 ± 7.2** | **85.3 ± 9.1** | |

ANOVA indicated no significant differences between distances for a given time point. Mortality was significantly higher at 48 hrs than 24 hrs (F(1, 12) = 24.15, P = 0.0004)

NOTE: Control (unsprayed but caged and exposed) mortality was 0% at 24 hours and 0.8% at 48 hours

significant differences in mean droplet size (VMD) among the three distances (ANOVA- F (2,6) = 0.3974, P = 0.6885). The VMD at 30.5 m, 70.0 m, and 91.4 m from the sprayer was 15.7 ± 1.5μ, 17.3 ± 3.8μ, and 15.3 ± 3.1μ, respectively (S5 Table). Further, no significant differences in droplet density were observed (S5 Table) across the spray field.

Mortality of these mixed site (Doral and Miami Beach), pyrethroid resistant *Cx. quinquefasciatus* was assessed at both 24 and 48 hours after treatment (Table 1). Mortality for untreated control locations was minimal at 24 and 48 hrs. The average mortality at 24 hours post-treatment for all 9 treatment sampling stations, regardless of distance, was 65.1 ± 7.2%. The average 48-hour mortality for all 9 treatment sampling stations was 85.3 ± 9.1%. No significant differences in mortality were observed between the individual distances at either 24 or 48 hours (F (2, 12) = 1.018, P = 0.3904). Mortality at 48 hours post-treatment was significantly greater than the mortality at 24 hours post-treatment (F(1, 12) = 24.15, P = 0.0004).

## Discussion

The insecticide susceptibility status of local mosquito populations is baseline information critical to the development and maintenance of an effective IVM program. Likewise, assessing new tools for adoption into IVM plans is also critical to maintain long-term treatment efficacy. The current study was conducted to help Miami-Dade County Mosquito Control maintain an effective IVM program by assessing the insecticide susceptibility of *Cx. quinquefasciatus* populations from 29 locations in Miami-Dade County, Florida, USA and to assess the ability of a new ULV space spray product to effectively control adulticide resistant Miami-Dade County *Cx. quinquefasciatus* mosquitoes.

Like in Miami-Dade County *Ae. aegypti* populations [14, 15], we found that pyrethroid insecticide resistance was widespread in Miami-Dade County *Cx. quinquefasciatus*. All *Cx. quinquefasciatus* populations displayed resistance to permethrin and deltamethrin with mortalities ranging from 0 to 76% at the DT. While molecular analyses demonstrated the 1014 F *kdr* mutation linked to pyrethroid insecticide resistance was present in every population we tested, organisms homozygous for the mutation were not detected in some locations which showed strong resistance. Unlike in *Ae. aegypti* where *kdr* can reach near 100%, no location, even the most resistant, had levels of the homozygous mutant that exceeded ~45% (Fig 3). Notably, heterozygous 1014F organisms made up large portions of the populations regardless of the level of phenotypic pyrethroid insecticide resistance observed in the CDC Bottle bioassay (Figs 1 and 3), which adds more support to studies that show the 1014F mutation is a smaller contributor to overall pyrethroid resistance in *Cx. quinquefasciatus* than enzymatic resistance [23, 25]. Here, we did not observe a strong positive correlation between *kdr*

genotype and insecticide susceptibility status except for a moderate negative correlation between the FF genotype and deltamethrin activity at the DT (S1 File).

While pyrethroid resistance was widespread in Miami-Dade County *Cx. quinquefasciatus*, organophosphate susceptibility varied. Numerous susceptible populations were observed for malathion and naled. Some populations, like those from Homestead and Florida City, were resistant to both classes of AI while others, like those from Doral & Sunset, were only resistant to pyrethroids. We suggest future studies using quantitative methods like topical application to produce insecticide median lethal doses for Miami-Dade County and susceptible *Cx. quinquefasciatus* would allow us to better define the intensity of the resistance and rigorously compare results between locations [14]. Characterization of enzymatic activity levels between Miami-Dade County populations may allow us to better understand the differences between populations that are resistant to both pyrethroids and OPs versus those resistant to only pyrethroids.

Observationally, we note that 6 of the 7 locations with resistance to naled were in the southwestern portion of the county, which tends to be more agricultural and less urban than the eastern and northern portions of Miami-Dade County. The potential correlation between agricultural insecticide usage and insecticide resistance in public health vectors has been suggested previously and deserves further investigation. Miles and Pfeuffer (1997) determined agriculture was the primary user of insecticides, followed by golf courses, and then domestic users and lastly mosquito control based on the residues in samples taken from South Florida canals [41]. In their study, the most common insecticides detected were those targeting the acetylcholinesterase system like methomyl (carbamate), ethoprophos (organophosphate), and phorate (organophosphate), which are used on vegetables and sugarcane. The organophosphate chlorpyrifos was also reported as the second-most used insecticide on domestic lawns and golf courses.

Although mosquito control is a minor contributor to the total quantity of insecticides used in Miami-Dade County [41], we identified intense resistance to pyrethroids and less intense but still widespread resistance to organophosphates. Taking into consideration that no aerial public health adulticide mission has been conducted in Miami-Dade County since 2017, our results showing more intense resistance (i.e., less mortality in bottles at 120 minutes) in both classes of AI in the southern portion of the county where agricultural activities are more common deserves further investigation. Insecticide use on private lawns, golf courses and for agriculture has often been reported as selective pressures that intensify pyrethroid resistance in insects [42–44]. Another interesting aspect of resistance development to consider, although studies in mosquitoes are limited, is that mosquito larvae often feed on plant debris or grow in water bodies enriched with defensive plant compounds and this may intensify resistance by enhancing insecticide tolerance in adult mosquito populations [45, 46]. South Florida has a climate favorable for growing crops and citrus production, and this agricultural pressure could be one reason for the intense resistance.

Regardless of the causes underlying insecticide resistance in *Cx. quinquefasciatus*, the mission to protect public health requires methods that are operationally effective. Just as in agricultural production operations, as resistance builds to classes of AIs, new products and methods need to be introduced to maintain efficacy [7]. New insecticides are urgently needed for effective adult control, especially during a mosquito-borne disease outbreak when immediate reduction in adult numbers is needed. Due to already strong resistance in the southern and western portions of Miami-Dade County to both pyrethroids and organophosphates, rotation between these two classes may not be enough to prevent the continued increase of insecticide resistance levels in Miami-Dade County *Cx. quinquefasciatus*.

Insecticide resistance observed in bioassays is not only a laboratory phenomenon but is also observed in field spray studies with *Cx. quinquefasciatus* populations and thus operationally

important. Recent Florida studies found an average 24 hr efficacy of approximately 54% for 5 different formulations while another showed one susceptible population and one population resistant (~50% mortality) to two formulations [25, 47]. Notably, both studies included some synergized products. Thus, we tested the newly EPA-approved multiple-AI product ReMoa Tri to see if it could provide an effective tool when adulticide use is required for control of resistant populations. We found the new adulticide product killed over 80% of the suburban, pyrethroid resistant Miami-Dade County *Cx. quinquefasciatus* by 48 hours post-treatment. The use of insecticides with new modes of action, or those with multiple classes of AI in locations where there is broad resistance (i.e. to both current classes of AI) in adult mosquito populations will help support resistance management and aid in the optimization of sustainable integrated vector control strategies. Multiple AI formulations are now being effectively used in long lasting insecticidal nets to control resistant malaria vectors [48, 49]. However, judicious use of these products is required to avoid the early selection and development of resistance to new AIs or combination products [50].

The plasticity of arthropods to adapt, or build resistance, to AIs of a few classes at the same time is possible although there appears to be a limit to this adaptation. Studies in predatory mites showed they could become resistant to three very different classes of AI at the same time but not to four [51]. Studies with the biolarvicide *Bacillus thuringiensis Israelensis* also seem to surpass this maximum adaptation threshold; even though strong resistance developed to individual toxins or a couple of toxins when pressured, little resistance developed to the full complement of 4–6 toxins in the natural mixture even after significant pressure [52, 53].

Finding effective strategies to manage mosquitoes remains a challenge, but strong IMM practices including the continual evaluation of new tools, re-evaluation of existing methods and approaches, and regular resistance testing will likely still be the most effective strategy for long-term sustainable mosquito abatement. We note that new products require concurrent development of methods to assess these products for comparison to existing products. As no published methods using ReMoa Tri in standard IR testing like CDC bottle bioassay, WHO tubes, or wind tunnel testing are available at this time, we had to conduct a field spray study to determine if this product was a potentially effective alternative for addition to the IMM program in Miami-Dade County. However, standard IR testing methodologies for new products are certainly needed as the time and expense involved in conducting field trials makes evaluation against multiple populations cost prohibitive. Standard testing methods for these new products and an understanding of how these IR testing assays relate to field performance would allow us to survey efficacy across a much larger area and against more populations.

We cautiously hope that new products with new or multiple modes of action like ReMoa Tri can give long-lasting adulticide options when immediate control of insecticide resistant populations is needed during outbreaks. Here, we observed 85% control using a new combination product in a field trail against our resistant Miami-Dade County *Cx. quinquefasciatus* and this provides a new tool to add to an IMM plan and assist in maintaining effective control.

## Supporting information

**S1 File. Data for CDC bottle bioaasay for Miami-Dade *Culex quinquefasciatus* populations tested and correlation analysis with kdr frequency.** This file contains a tab for each collection with the pesticide susceptibility data and an additional tab with the results of the correlation analysis.
(XLSX)

**S1 Table. *Cx. quinquefasciatus* collection information.**
(DOCX)

**S2 Table. Miami-Dade County *Cx. quinquefasciatus* mortality at diagnostic times and 120 minutes of exposure in CDC bottle bioassays against multiple active ingredients.**
(DOCX)

**S3 Table. Genotype and Allele Percentages for the leucine to phenylalanine 1014 mutation in Miami-Dade county *Cx. quinquefasciatus* populations.**
(DOCX)

**S4 Table. Weather during ReMoa Tri ground ULV field trial.**
(DOCX)

**S5 Table. Droplet parameters from ReMoa Tri ground ULV field trial.**
(DOCX)

## Acknowledgments

The contents of this manuscript are solely the responsibility of the authors and do not necessarily represent the official views of the Centers for Disease Control and Prevention, the Department of Health and Human Services, the USDA, the State of Florida, or Miami-Dade County. Mention of trade names or commercial products in this publication is solely for the purpose of providing specific information and does not imply recommendation or endorsement by the U.S. Department of Agriculture or any governmental body. USDA is an equal opportunity provider and employer.

The authors thank the members of Miami-Dade Mosquito Control, Maday Moreno, Pedro Errasti, Joseph Blackman, Darrell Cochran, Carlos Lanzas and Yanet Chiong.

## Author Contributions

**Conceptualization:** Isik Unlu, Eva A. Buckner, Chalmers Vasquez, Alden S. Estep.

**Data curation:** Isik Unlu, Eva A. Buckner, Johanna Medina, Aimee Cabrera, Ana L. Romero-Weaver, Daviela Ramirez, Kyle J. Kosinski, T. J. Fedirko, Leigh Ketelsen, Chelsea Dorsainvil, Alden S. Estep.

**Formal analysis:** Isik Unlu, Natalie L. Kendziorski, Alden S. Estep.

**Funding acquisition:** Eva A. Buckner, Alden S. Estep.

**Investigation:** Isik Unlu, Eva A. Buckner, Johanna Medina, Aimee Cabrera, Ana L. Romero-Weaver, Daviela Ramirez, Natalie L. Kendziorski, Kyle J. Kosinski, T. J. Fedirko, Leigh Ketelsen, Chelsea Dorsainvil, Alden S. Estep.

**Project administration:** Isik Unlu, Alden S. Estep.

**Resources:** Eva A. Buckner, Johanna Medina, Chalmers Vasquez, Aimee Cabrera, Alden S. Estep.

**Supervision:** Isik Unlu, Chalmers Vasquez.

**Writing – original draft:** Isik Unlu, Alden S. Estep.

**Writing – review & editing:** Isik Unlu, Eva A. Buckner, Alden S. Estep.

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
