## [Decision Letter · Decision Letter 0]

15 Sep 2023

PONE-D-23-21970Crouching Tiger, Obvious Trouble: Insecticide resistance of Miami-Dade Culex quinquefasciatus populations and initial field efficacy of a new resistance-breaking adulticide formulationPLOS ONE

Dear Dr. Estep,

Thank you for submitting your manuscript to PLOS ONE. After careful consideration, we feel that it has merit but does not fully meet PLOS ONE’s publication criteria as it currently stands. Therefore, we invite you to submit a revised version of the manuscript that addresses the points raised during the review process.

We look forward to receiving your revised manuscript.

Kind regards,

Junaid Rahim, Ph.D

Academic Editor

PLOS ONE

Journal Requirements:

4. We note that you have a patent relating to material pertinent to this article:

"I have read the journal's policy and the authors of this manuscript have the following competing interests: ReMoa Tri for field testing was provided by Valent Biosciences LLC under an Experimental Use Permit."

Please provide an amended statement of Competing Interests to declare this patent (with details including name and number), along with any other relevant declarations relating to employment, consultancy, patents, products in development or modified products etc. Please confirm that this does not alter your adherence to all PLOS ONE policies on sharing data and materials, as detailed online in our guide for authors http://journals.plos.org/plosone/s/competing-interests by including the following statement: "This does not alter our adherence to  PLOS ONE policies on sharing data and materials.” 

If there are restrictions on sharing of data and/or materials, please state these. Please note that we cannot proceed with consideration of your article until this information has been declared.

**Additional Editor Comments:** Authors are requested to incorporate the suggestions/queries raised by the reviewers. My suggestion is to change the existing title to:Insecticide resistance of Miami-Dade Culex quinquefasciatus populations and initial field efficacy of a new resistance-breaking adulticide formulation

Reviewers' comments:

Reviewer's Responses to Questions

**Comments to the Author**

1. Is the manuscript technically sound, and do the data support the conclusions?

Reviewer #1: Yes

Reviewer #2: Yes

2. Has the statistical analysis been performed appropriately and rigorously? 

Reviewer #1: No

Reviewer #2: Yes

3. Have the authors made all data underlying the findings in their manuscript fully available?

Reviewer #1: Yes

Reviewer #2: Yes

4. Is the manuscript presented in an intelligible fashion and written in standard English?

Reviewer #1: Yes

Reviewer #2: Yes

5. Review Comments to the Author

Reviewer #1: The study titled "Crouching Tiger, Obvious Trouble: Insecticide Resistance of Miami-Dade Culex quinquefasciatus Populations and Initial Field Efficacy of a New Resistance-Breaking Adulticide Formulation" by Unlu et al., is technically sound, well-focused, methodologically accurate, and well-written. However, below are my concerns and comments for improvement of the paper.

The authors collected mosquitoes from two different areas: one treated with adulticides year-round and the other treated only during peak mosquito season. However, this distinction wasn't fully explored in the analysis, results, and discussion. I think a great aspect of the work is missing.

Methods have been presented in too much detail. To present concisely, avoid listing the details of the experiments that are widely used or already published in CDC guidelines and numerous articles.

Presentation of Table 1 could be improved by using geographical maps for better visualization. The data in Table 1 might be more suitable for supplementary material.

Similarly, Table 2's complexity could be addressed by using a line diagram with time on the x-axis and percentage mortality on the y-axis.

Figures 2 and 3, which illustrate insecticide resistance in mosquitoes to different classes of insecticides, could potentially be superimposed and shown using different symbols for different insecticides. Doing that a reader could have compared the differences of each insecticide. However, this suggestion as optional.

Table 3: present this in a bar diagram to display bi-locus and tri-locus frequencies.

Discussion: “a statement about the highest level of organophosphate in the southwest portion” …. this interpretation could be statistically supported through a heat-map analysis instead of a border interpretation, which may not be correct as well.

In summary, this is a great and huge amount of work for which I appreciate researchers. The "ReMoa Tri ground ULV field trial" is impressive. I endorse the revised version of this article for publication.

Reviewer #2: The work by Unlu and colleagues report insecticide resistance in several populations of Culex quince in Florida, USA and show susceptibility of those resistant mosquitoes to a new insecticide, ReMoa, that targets the adult mosquito stage. The study designs were appropriate and effective. The study outcomes will be of high value to vector control workers and may be of interest to a broader readership. My comments below are suggestions that I believe may help improve how readers digest the study results and point to one limitation of the manuscript.

General Comments to Authors:

The authors should probably include an Ethics Statement since this study employed field research.

Conducting bottle bioassays (BBA) using ReMoa so that they could be compared to those they report for permethrin and resmethrin would substantially improve the study and increase its broader impact. Doing a BBA for ReMoa is an obvious step; it’s absence would be noted by any reader. This reviewer is left wondering whether if its absence was due to ReoMoa being less effective in BBA compared to conventional insecticides such as permethrin and detamethrin. Given the disclosed conflicts of interest, there is a risk to the authors that readers will see their research as overly influenced by financial considerations or industry connections. That stated, a BBA for ReMoa could be perceived as beyond the scope of the study design, and not necessary. But if the data is available, this reviewer highly recommends it be included in the manuscript.

Comments by line number or Table / Figure:

33 and elseswhere: Authors indicate fractions for resistance to pyrethroids and others (e.g., 2 of 27+28) but provide proportions for mortality data with the ReaMoa field trial. To improve comparisons, please provide proportions (i.e, percent of total) rather than fractions throughout the manuscript.

77: Authors provide Figure 1 in the introduction that shows arbovirus disease incidence for the study region as reported by the health department. However, there are no methods or analysis of substance to support the data that is presented in Figure 1. Additionally, since these data are related to humans, there may be a need to provide an ethics statement related to it. Recommend data in Fig 1 be moved to Results and the methods described, or it be removed and instead referenced to a report or data set from the health department.

85 and 87: Please provide a supporting reference for each sentence.

Table 1: The data in this table is presented in a manner that is complex and too difficult for a reader to analyze without running it through other software. I recommend that the data in Table 1 be summarized as a bar graph and Table 1 be moved to Supplementary Information. I do not recommend each row be an individual bar in the graph, rather generate summary statistics and/or group geographic regions (e.g., authors could combine data from study sites that are near to each other, or combine using a political boundary (e.g., city, or other division. The preceding are merely suggestions that come to my mind; I’m sure there are other reasonable approaches)).

134-253: This section of the methods provides more detail than is necessary (it reads more like a protocol for which the step numbers have been removed rather than a methods section of a manuscript). Consider condensing this and/or moving the details to Supplementary Information.

166-182: Similar to above, more detail is provided than is necessary, particularly if the Methods are identical to the referenced CDC article. If so, I recommend stating that the BBA was conducted as described in the CDC article.

187-190: This appears to belong in the Results section.

198-200: Recommend removing as it’s not relevant to the study (it reads more like additional steps in a protocol).

267: State how many hours after sunrise rather than the time of day.

281: Provide manufacturer information for DropVision. Do similar if other manufacturer attributions are missing.

289-295: This appears to belong in the Methods

Tables 2 and 3: Similar to Table 1, the information here should be summarized as a graph so that it can be more easily understood (the table can be moved to Supplementary Information).

Table 3 Footnote 1: This should be moved to Methods.

371: If the Cx quince used in the ReMoa trial were resistant to pyrethroids, please emphasize that here.

Table 6: What do A and B refer to in the final row (Avg. mortality)?

409: “some populations” are noted. Please indicate which are the “some”

410: Please explain what a topical application assay is or provide a reference that does.

452: As noted above, the lack of BBA data for ReMoa is a limitation of the study and should be considered. Particularly since the 24 h mortality for ReoMoa (65%; Table 6) was lower than the highest mortality for the pyrethroids (range of 0 – 76% at the DT of 30-60 min; line 395). This should be explained in the Discussion or Conclusion.

References: Please check formatting (e.g., itallics for species (line 537 and elsewhere).

6. PLOS authors have the option to publish the peer review history of their article (what does this mean?). If published, this will include your full peer review and any attached files.

Reviewer #1: No

Reviewer #2: No

---

## [Author Response · Author response to Decision Letter 0]

2 Nov 2023

Editorial Comments on Journal Requirements:

Author response: This revision has been checked for formatting. Title, author affiliations, file naming and reference formatting have been substantially revised per the guidance above.

Author response: No permits were required for surveillance collections. Property owner permission was sought and provided as needed. This is now addressed in text in lines 120-124 of the revised manuscript.

Author response: This is addressed in the resubmission and resubmission cover letter.

4. We note that you have a patent relating to material pertinent to this article:

"I have read the journal's policy and the authors of this manuscript have the following competing interests: ReMoa Tri for field testing was provided by Valent Biosciences LLC under an Experimental Use Permit."

Please provide an amended statement of Competing Interests to declare this patent (with details including name and number), along with any other relevant declarations relating to employment, consultancy, patents, products in development or modified products etc. Please confirm that this does not alter your adherence to all PLOS ONE policies on sharing data and materials, as detailed online in our guide for authors http://journals.plos.org/plosone/s/competing-interests by including the following statement: "This does not alter our adherence to PLOS ONE policies on sharing data and materials.” 

If there are restrictions on sharing of data and/or materials, please state these. Please note that we cannot proceed with consideration of your article until this information has been declared.

Author response: Please note that we cited the Valent patent in both the original and revised manuscript as [34]. None of the Authors of this manuscript have a patent related to ReMoa Tri or are any of the Authors employees of Valent Biosciences LLC. None of the Authors have a financial relationship with Valent Biosciences LLC. ReMoa Tri is now fully EPA approved and available for anyone with legal status to purchase so there are no issues with obtaining the formulation. All data is shared within the manuscript.

We have confirmed as requested and included the above statement in our resubmission letter.

Author response: We have carefully reviewed and reformatted the references. We have checked online versions of each citation and did not find any retracted articles. Notably, there were two articles that had published errata which we have now included in the citations per the ICJME reference style cited by the PLoS One format guide. For [15], the erratum fixed 2 figure captions that had been reversed. For [36], the erratum concerned 22 species of Ochlerotatus that had been overlooked. Neither errata was material to our use of the reference so we choose to keep both.

Additional Editor Comments:

Authors are requested to incorporate the suggestions/queries raised by the reviewers. My suggestion is to change the existing title to:Insecticide resistance of Miami-Dade Culex quinquefasciatus populations and initial field efficacy of a new resistance-breaking adulticide formulation

Author response: Title changed as requested. 

 Reviewer #1: The study titled "Crouching Tiger, Obvious Trouble: Insecticide Resistance of Miami-Dade Culex quinquefasciatus Populations and Initial Field Efficacy of a New Resistance-Breaking Adulticide Formulation" by Unlu et al., is technically sound, well-focused, methodologically accurate, and well-written. However, below are my concerns and comments for improvement of the paper.

Author response: Thank you. We appreciate the effort you put into comments for improving the manuscript and we have incorporated most of them.

The authors collected mosquitoes from two different areas: one treated with adulticides year-round and the other treated only during peak mosquito season. However, this distinction wasn't fully explored in the analysis, results, and discussion. I think a great aspect of the work is missing.

Author response: All samples were collected from areas that are subject to full year control, so it is not possible to conduct the analysis requested by the Reviewer. We do see the source of the confusion (existing lines 129-131: ” We chose areas that are treated with adulticides year-round versus areas treated only during peak mosquito season (June-November)” and have revised the text to be more clear that we only sampled areas with year round control operations.

Methods have been presented in too much detail. To present concisely, avoid listing the details of the experiments that are widely used or already published in CDC guidelines and numerous articles.

Presentation of Table 1 could be improved by using geographical maps for better visualization. The data in Table 1 might be more suitable for supplementary material.

Similarly, Table 2's complexity could be addressed by using a line diagram with time on the x-axis and percentage mortality on the y-axis.

Author response: In response to the suggestion of the reviewer we have reduced the detail for the bottle bioassay, the kdr assay, field spray and field collections methods. 

The sample collection metadata shown in Table 1 is critical information and we feel should be kept in tabular form. We note that subsequent usage of these locations is in graphical form in Figures 2 & 3. 

We do agree with the Reviewer with respect to Table 2 and have moved this information into a supplemental file since the four columns containing the results at the diagnostic time are presented visually in Figures 2 & 3.

Figures 2 and 3, which illustrate insecticide resistance in mosquitoes to different classes of insecticides, could potentially be superimposed and shown using different symbols for different insecticides. Doing that a reader could have compared the differences of each insecticide. However, this suggestion as optional.

Author response: We considered this suggestion but think superimposing the mortality for multiple active ingredients onto the same map would be difficult to do clearly. The more intensely sampled northeastern portion of the county was already difficult to annotate for a single AI so adding a second AI would be visually unacceptable.

Table 3: present this in a bar diagram to display bi-locus and tri-locus frequencies.

Author response: This manuscript only assesses the frequency and heterozygosity of a SNP at a single locus. While we are not sure of the bi- and tri-locus frequencies to which the Reviewer refers, we have placed this information in as a supplemental table since it is presented graphically in pie charts in Figure 3.

Discussion: “a statement about the highest level of organophosphate in the southwest portion” …. this interpretation could be statistically supported through a heat-map analysis instead of a border interpretation, which may not be correct as well.

Author response: We have considered the suggestion but wish to leave this issue unexplored in this manuscript because we do not feel that CDC Bottle bioassay data has the rigor needed for a valid statistical analysis that would underpin a solid heatmap. CDC assay data is unreplicated by definition as the bottles are not independent. We do think that the reviewer suggestion is a good one and as part of the discussion call for a future study more rigorous quantification of resistance intensity by topical application that would allow such an analysis.

In summary, this is a great and huge amount of work for which I appreciate researchers. The "ReMoa Tri ground ULV field trial" is impressive. I endorse the revised version of this article for publication.

Author response: Thank you for the kind comments and the suggestions for improving this manuscript.

Reviewer #2: The work by Unlu and colleagues report insecticide resistance in several populations of Culex quince in Florida, USA and show susceptibility of those resistant mosquitoes to a new insecticide, ReMoa, that targets the adult mosquito stage. The study designs were appropriate and effective. The study outcomes will be of high value to vector control workers and may be of interest to a broader readership. My comments below are suggestions that I believe may help improve how readers digest the study results and point to one limitation of the manuscript.

Author response: Thank you for the effort in reviewing this manuscript and the well-considered suggestions for revision. We have incorporated most of them. We think the result is a better manuscript.

General Comments to Authors:

The authors should probably include an Ethics Statement since this study employed field research.

Conducting bottle bioassays (BBA) using ReMoa so that they could be compared to those they report for permethrin and resmethrin would substantially improve the study and increase its broader impact. Doing a BBA for ReMoa is an obvious step; it’s absence would be noted by any reader. This reviewer is left wondering whether if its absence was due to ReoMoa being less effective in BBA compared to conventional insecticides such as permethrin and detamethrin. Given the disclosed conflicts of interest, there is a risk to the authors that readers will see their research as overly influenced by financial considerations or industry connections. That stated, a BBA for ReMoa could be perceived as beyond the scope of the study design, and not necessary. But if the data is available, this reviewer highly recommends it be included in the manuscript.

Author response: We agree with the Reviewer that a ReMoa Tri BBA would be a useful tool but to our knowledge no method for this bioassay has been published. We also agree that such an assay may be less than impressive when used within the standard timeframe of the BBA due to the slower acting AIs. However, development of such an assay is beyond the scope of the work presented. Our access to and use of ReMoa Tri was limited to the field formulation provided by Valent BioSciences LLC under their EUP thus no BBA data is available.

Comments by line number or Table / Figure:

33 and elseswhere: Authors indicate fractions for resistance to pyrethroids and others (e.g., 2 of 27+28) but provide proportions for mortality data with the ReaMoa field trial. To improve comparisons, please provide proportions (i.e, percent of total) rather than fractions throughout the manuscript.

Author response: We have revised the text to be consistent.

77: Authors provide Figure 1 in the introduction that shows arbovirus disease incidence for the study region as reported by the health department. However, there are no methods or analysis of substance to support the data that is presented in Figure 1. Additionally, since these data are related to humans, there may be a need to provide an ethics statement related to it. Recommend data in Fig 1 be moved to Results and the methods described, or it be removed and instead referenced to a report or data set from the health department.

Author response: Figure 1 was created from publicly available data published by the FL Department of Health and is now cited as such by reference as suggested by the Reviewer. We have removed figure 1 from the manuscript and just leave the text information in place with citation.

85 and 87: Please provide a supporting reference for each sentence.

Author response: Added as suggested.

Table 1: The data in this table is presented in a manner that is complex and too difficult for a reader to analyze without running it through other software. I recommend that the data in Table 1 be summarized as a bar graph and Table 1 be moved to Supplementary Information. I do not recommend each row be an individual bar in the graph, rather generate summary statistics and/or group geographic regions (e.g., authors could combine data from study sites that are near to each other, or combine using a political boundary (e.g., city, or other division. The preceding are merely suggestions that come to my mind; I’m sure there are other reasonable approaches)).

Author response: Table 1 is the sample collection data and is thus critical information that we feel should be present in the body of the manuscript. Graphical representation of the location information is the basis for the points in Figures 1-3 and it is unclear to us how we can present location information as a bar graph as suggested. If the Reviewer means Table 2, which presents the bioassay data, we agree and have turned Table 2 into a supplemental table. The diagnostic time bioassay data for the four AIs is presented graphically as suggested in Figures 1 & 2.

134-253: This section of the methods provides more detail than is necessary (it reads more like a protocol for which the step numbers have been removed rather than a methods section of a manuscript). Consider condensing this and/or moving the details to Supplementary Information.

Author response: The methods have been significantly revised to be less detailed.

166-182: Similar to above, more detail is provided than is necessary, particularly if the Methods are identical to the referenced CDC article. If so, I recommend stating that the BBA was conducted as described in the CDC article.

Author response: The methods have been revised to be less detailed.

187-190: This appears to belong in the Results section.

Author response: Text revised.

198-200: Recommend removing as it’s not relevant to the study (it reads more like additional steps in a protocol).

Author response: Text revised as suggested.

267: State how many hours after sunrise rather than the time of day.

Author response: Text revised as suggested.

281: Provide manufacturer information for DropVision. Do similar if other manufacturer attributions are missing.

Author response: Text revised as suggested.

289-295: This appears to belong in the Methods

Author response: Text revised as suggested.

Tables 2 and 3: Similar to Table 1, the information here should be summarized as a graph so that it can be more easily understood (the table can be moved to Supplementary Information).Author response: Tables 2 & 3 have been placed as supplemental information as requested.

Table 3 Footnote 1: This should be moved to Methods.

Author response: Text revised as suggested.

371: If the Cx quince used in the ReMoa trial were resistant to pyrethroids, please emphasize that here.

Author response: Text revised as suggested.

Table 6: What do A and B refer to in the final row (Avg. mortality)?

Author response: The A & B are removed.

409: “some populations” are noted. Please indicate which are the “some”

Author response: Text revised as suggested.

410: Please explain what a topical application assay is or provide a reference that does.

Author response: Reference added.

452: As noted above, the lack of BBA data for ReMoa is a limitation of the study and should be considered. Particularly since the 24 h mortality for ReoMoa (65%; Table 6) was lower than the highest mortality for the pyrethroids (range of 0 – 76% at the DT of 30-60 min; line 395). This should be explained in the Discussion or Conclusion.

Author response: Per the suggestion of the Reviewer, we have added discussion noting the lack of standard resistance testing methods like BBA or WHO tubes for ReMoa Tri. This is found in lines 420-429. Note that we do not want to speculate about possible BBA outcomes since this was not part of our study nor do we want to speculate on thresholds between BBA data and field trial data since there is no data for us to make an informed statement. However, we agree that a method for standardized IR testing needs to be available when a new product hits the market and we call for such. Having to rely only on time consuming field trials is not a widely useful method for the many large mosquito control programs that exist.

References: Please check formatting (e.g., itallics for species (line 537 and elsewhere).

Author response: References have been checked and reformatted.

---

## [Decision Letter · Decision Letter 1]

1 Dec 2023

PONE-D-23-21970R1Insecticide resistance of Miami-Dade Culex quinquefasciatus populations and initial field efficacy of a new resistance-breaking adulticide formulationPLOS ONE

Dear Dr. Estep,

Thank you for submitting your manuscript to PLOS ONE. After careful consideration, we feel that it has merit but does not fully meet PLOS ONE’s publication criteria as it currently stands. Therefore, we invite you to submit a revised version of the manuscript that addresses the points raised during the review process.

**Authors are requested to revise the MS according to the suggestions given by reviewer 2.**

We look forward to receiving your revised manuscript.

Kind regards,

Junaid Rahim, Ph.D

Academic Editor

PLOS ONE

Journal Requirements:

Reviewers' comments:

Reviewer's Responses to Questions

**Comments to the Author**

1. If the authors have adequately addressed your comments raised in a previous round of review and you feel that this manuscript is now acceptable for publication, you may indicate that here to bypass the “Comments to the Author” section, enter your conflict of interest statement in the “Confidential to Editor” section, and submit your "Accept" recommendation.

Reviewer #1: All comments have been addressed

Reviewer #2: (No Response)

2. Is the manuscript technically sound, and do the data support the conclusions?

Reviewer #1: Yes

Reviewer #2: Yes

3. Has the statistical analysis been performed appropriately and rigorously? 

Reviewer #1: Yes

Reviewer #2: Yes

4. Have the authors made all data underlying the findings in their manuscript fully available?

Reviewer #1: Yes

Reviewer #2: Yes

5. Is the manuscript presented in an intelligible fashion and written in standard English?

Reviewer #1: Yes

Reviewer #2: Yes

6. Review Comments to the Author

Reviewer #1: The study designs, amount of work and paper writing are appropriate and effective. The result of the study is of high value to vector control workers. Paper reads nice. Overall, this is a nice piece of work, and should be published.

Reviewer #2: Thank you for addressing the reviewer comments. There are a few minor edits that I'm recommending that are indicated below.

Including Table 1 in the body of the manuscript rather supplementary information remains questionable to me. It's not a measurement, and while the information is essential for the study, it can be looked up in supplementary information for readers that want those details. Additionally, the sites are indicated on the maps of Figure 1 and 2. I think moving it to supplementary is reasonable, but I can see authors perspective that moving it results in 3 figures and 1 table in the body of the manuscript.

Line 147: please indicate here if CMAVE strain is susceptible or resistant, and to which insecticide class.

206: remove additional "."

249: this sentence is grammatically incorrect, and doesn't make sense as written. Please rephrase.

319 - 323: Data here repeats verbatim what is shown in Table 2. To avoid repetition, consider converting Table 2 to a histogram (distance on the x- axis, with bars side-by-side for 24 h and 48h time points, y-axis as % mortality). Alternatively, consider summarizing the data in the body of the manuscript, and keep the Table 2 as is. I don't consider this essential, but most journals refrain from showing the same data twice.

348: inset space between "Figs 1"

398: correct capitalization and punctuation.

401 - 402 and 420 - 429: The following is optional for the authors to consider if they plan to submit a revised manuscript. I encouraged the authors to consider including bottle bioassays (BBA) for ReMoa Tri in hopes it would provide solid data that directly compares ReMoa Tri to insecticides that are used currently. While I appreciate the authors explanation for their absence is that there is currently no CDC or WHO BBA for ReMoa Tri, readers may find that explanation unsatisfying as BBA can be conducted with products that are applied in the field. While this isn't encouraged by CDC or WHO for current insecticide classes, ReMoa is a proprietary mix of active ingredients that are unlikely to be easily sourced separately any time soon for reconstitution in CDC-like BBA that utilizes technical grade ingredients (e.g., like permethrin synergized with PBO). In the absence of ReMoa Tri BBA, the authors should consider comparing the results of the ULV field application trials with ReMoa Tri to published ULV studies with permethrin, deltamethrin, malation, and/or naled. While mosquito populations in this and published studies are not identical, there are some that were done in areas with known insecticide resistance, and could be used to provide an approximate comparison of ReMoa Tri to other insecticides that are widely used. This recommendation is not to make preparing a revision unwieldy (and it is optional for authors to consider), but is aimed at helping the authors to provide a balanced analysis of ReMoa Tri vs current insecticide classes. ReMoa Tri seems to be a good product, but readers may be left wondering how it compares to the chemistries that are already on the market. A discussion of this could benefit the manuscript, but again, not required in my opinion.

7. PLOS authors have the option to publish the peer review history of their article (what does this mean?). If published, this will include your full peer review and any attached files.

Reviewer #1: No

Reviewer #2: No

---

## [Author Response · Author response to Decision Letter 1]

3 Dec 2023

The same information below is in our Response to Reviewers file:

Reviewer #1: The study designs, amount of work and paper writing are appropriate and effective. The result of the study is of high value to vector control workers. Paper reads nice. Overall, this is a nice piece of work, and should be published.

Reviewer #2: Thank you for addressing the reviewer comments. There are a few minor edits that I'm recommending that are indicated below.

Including Table 1 in the body of the manuscript rather supplementary information remains questionable to me. It's not a measurement, and while the information is essential for the study, it can be looked up in supplementary information for readers that want those details. Additionally, the sites are indicated on the maps of Figure 1 and 2. I think moving it to supplementary is reasonable, but I can see authors perspective that moving it results in 3 figures and 1 table in the body of the manuscript.

Author Response: Considering the Reviewer comments, we have removed Table 1 and placed it in as a supplemental table. The text has been edited to accommodate this change.

Line 147: please indicate here if CMAVE strain is susceptible or resistant, and to which insecticide class. Author Response: The CMAVE strain is a laboratory susceptible strain and this is stated in Line 148. The tox profile of this strain has been widely examined so we have added a citation.

206: remove additional "." Author Response: Accepted

249: this sentence is grammatically incorrect, and doesn't make sense as written. Please rephrase. Author Response: Sentence revised. 

319 - 323: Data here repeats verbatim what is shown in Table 2. To avoid repetition, consider converting Table 2 to a histogram (distance on the x- axis, with bars side-by-side for 24 h and 48h time points, y-axis as % mortality). Alternatively, consider summarizing the data in the body of the manuscript, and keep the Table 2 as is. I don't consider this essential, but most journals refrain from showing the same data twice.

Author Response: This section has been revised considering the Reviewer comments. Rather than convert the table to a figure, we have instead revised the text. The text now gives a summary of the findings and the important statistical result rather than restating all the data.

348: inset space between "Figs 1" Author Response: Accepted

398: correct capitalization and punctuation. Author Response: Corrected

401 - 402 and 420 - 429: The following is optional for the authors to consider if they plan to submit a revised manuscript. I encouraged the authors to consider including bottle bioassays (BBA) for ReMoa Tri in hopes it would provide solid data that directly compares ReMoa Tri to insecticides that are used currently. While I appreciate the authors explanation for their absence is that there is currently no CDC or WHO BBA for ReMoa Tri, readers may find that explanation unsatisfying as BBA can be conducted with products that are applied in the field. While this isn't encouraged by CDC or WHO for current insecticide classes, ReMoa is a proprietary mix of active ingredients that are unlikely to be easily sourced separately any time soon for reconstitution in CDC-like BBA that utilizes technical grade ingredients (e.g., like permethrin synergized with PBO). In the absence of ReMoa Tri BBA, the authors should consider comparing the results of the ULV field application trials with ReMoa Tri to published ULV studies with permethrin, deltamethrin, malation, and/or naled. While mosquito populations in this and published studies are not identical, there are some that were done in areas with known insecticide resistance, and could be used to provide an approximate comparison of ReMoa Tri to other insecticides that are widely used. This recommendation is not to make preparing a revision unwieldy (and it is optional for authors to consider), but is aimed at helping the authors to provide a balanced analysis of ReMoa Tri vs current insecticide classes. ReMoa Tri seems to be a good product, but readers may be left wondering how it compares to the chemistries that are already on the market. A discussion of this could benefit the manuscript, but again, not required in my opinion.

Author Response: We must reiterate that developing a BBA for this product was beyond the scope of the work that we undertook. First, the BBAs conducted on most of the collections in this study occurred before the ReMoa Tri formulation was available so it would have been quite impossible to test these populations with the product. Second, we did not have access to the formulation except for the field study under the specific terms of the EPA’s EUP.

As we noted in our previous response to reviewers about this issue at the first revision, we agree with this Reviewer that a bottle bioassay method is sorely needed but we exercise the option (given here by the Reviewer), not to include one in this study. It is an area for future work, and we indicate this in the manuscript (Lines 425-431).

We do also agree that we should have included a bit more context, so we have edited the text and cited a couple of recent field spray studies with Florida Culex populations to show that the IR seen in the lab is also observed in the field with commercial formulations. Even though these populations are not specifically those we tested, they do demonstrate that IR exists in many field populations to current formulations and give some baseline of how current formulations perform.

---

## [Editor Report · Decision Letter 2]

5 Dec 2023

Insecticide resistance of Miami-Dade Culex quinquefasciatus populations and initial field efficacy of a new resistance-breaking adulticide formulation

PONE-D-23-21970R2

Dear Dr. Estep,

We’re pleased to inform you that your manuscript has been judged scientifically suitable for publication and will be formally accepted for publication once it meets all outstanding technical requirements.

Kind regards,

Junaid Rahim, Ph.D

Academic Editor

PLOS ONE
---

## [Editor Report · Acceptance letter]

11 Dec 2023

PONE-D-23-21970R2 

Insecticide resistance of Miami-Dade *Culex quinquefasciatus* populations and initial field efficacy of a new resistance-breaking adulticide formulation 

Dear Dr. Estep:

I'm pleased to inform you that your manuscript has been deemed suitable for publication in PLOS ONE. Congratulations! Your manuscript is now with our production department. 

Kind regards, 

on behalf of

Dr. Junaid Rahim 

Academic Editor

PLOS ONE